# Hydroxylation of the Acetyltransferase NAA10 Trp38 Is Not an Enzyme-Switch in Human Cells

**DOI:** 10.3390/ijms222111805

**Published:** 2021-10-30

**Authors:** Rasmus Ree, Karoline Krogstad, Nina McTiernan, Magnus E. Jakobsson, Thomas Arnesen

**Affiliations:** 1Department of Biomedicine, University of Bergen, Jonas Lies vei 91, 5009 Bergen, Norway; karo.krogstad@hotmail.com (K.K.); nina.tiernan@uib.no (N.M.); 2Department of Immunotechnology, Lund University, Medicon Village, 22100 Lund, Sweden; magnus.jakobsson@immun.lth.se; 3Department of Biosciences, University of Bergen, 5009 Bergen, Norway; 4Department of Surgery, Haukeland University Hospital, 5021 Bergen, Norway

**Keywords:** protein hydroxylation, protein acetylation, proteomics, POST-translational modification, *N*-terminal acetyltransferase, NAA10

## Abstract

NAA10 is a major *N*-terminal acetyltransferase (NAT) that catalyzes the cotranslational *N*-terminal (Nt-) acetylation of 40% of the human proteome. Several reports of lysine acetyltransferase (KAT) activity by NAA10 exist, but others have not been able to find any NAA10-derived KAT activity, the latter of which is supported by structural studies. The KAT activity of NAA10 towards hypoxia-inducible factor 1α (HIF-1α) was recently found to depend on the hydroxylation at Trp38 of NAA10 by factor inhibiting HIF-1α (FIH). In contrast, we could not detect hydroxylation of Trp38 of NAA10 in several human cell lines and found no evidence that NAA10 interacts with or is regulated by FIH. Our data suggest that NAA10 Trp38 hydroxylation is not a switch in human cells and that it alters its catalytic activity from a NAT to a KAT.

## 1. Introduction

Protein *N*-terminal acetylation (Nt-acetylation) is pervasive and highly conserved among eukaryotes. Co- or posttranslational Nt-acetylation occurs on approximately 90% of all human proteins [1]. A functional effect of Nt-acetylation on a substrate protein has been described for a modest number of substrates. These effects include the regulation of folding [2,3,4,5,6], shortening or extending protein half-life [7,8,9,10,11], complex formation [12,13,14,15], and subcellular targeting [16,17,18]. Moreover, Nt-acetylation participates in crosstalk with serine phosphorylation and arginine methylation at the *N*-terminus of histone H4 [19,20,21,22].

*N*-terminal acetyltransferases (NATs) catalyze Nt-acetylation. Currently, seven human NATs have been discovered. They have been termed NatA-NatF and NatH and have catalytic subunits NAA10-NAA60 and NAA80 [1]. The catalytic subunits bind acetyl-coenzyme A (Ac-CoA) and catalyze the transfer of an acetyl group from Ac-CoA to the *N*-terminal amino group of the protein substrate. NatA [23], NatB [24,25], NatC [24], and NatE [23,24] also have auxiliary subunits that anchor the catalytic subunit to the ribosome, while NAA40 is presumed to bind the ribosome directly [1,26,27]. NatG and a group of plant-specific GNAT enzymes with dual NAT and KAT activity localizes to the chloroplasts [28,29]. NatF, which is localized to the Golgi apparatus and facesthe cytosol [30,31], and NatH, the actin NAT [32,33,34,35,36], are not ribosome-bound and have posttranslational activities.

The seven human NATs are all members of the GNAT superfamily [1]. They contribute to the Nt-acetylome to varying degrees and are directed by the *N*-terminal residues of the substrate protein according to their distinct substrate specificities. Chief among them in terms of substrate number is NatA, which acetylates an estimated 40% of the human proteome [37]. NatA is composed of the catalytic subunit NAA10, the auxiliary, ribosome-binding subunit NAA15 [23,38,39,40], and the small regulatory protein HYPK [5,41]. In addition, NAA50, the catalytic subunit of the NatE complex, is associated with NAA15 and NAA10 [23]. NatA may Nt-acetylate proteins with Ala, Cys, Gly, Ser, Thr, or Val in the second position after the initiator methionine has been removed by one of two methionine aminopeptidases [1,37,42,43]. NAA10 also has NAT activity that is independent of the NatA complex and that prefers acidic *N*-termini [43], although it is unclear to what extent such *N*-termini are actually in vivo substrates since the most abundant human proteins of this type, the actins, are modified by NatH [32]. In addition to its NAT activities, acetyltransferase-independent NAA10 interaction with DNA methyltransferase 1 (DNMT1) facilitates DNA methylation and the maintenance of genomic imprinting [44] as well as tumor suppressor silencing [45].

In addition to its well-established role as a NAT, several articles have proposed that NAA10 regulates diverse cellular processes by acetylating lysines in various proteins thus acting as a KAT [46,47,48,49,50,51,52,53,54]. Among these, hypoxia-inducible factor 1α (HIF-1α) has been reported to be acetylated on Lys532 [52]. This acetylation was proposed to render HIF-1α more unstable under normoxic conditions by facilitating interaction with the von Hippel-Lindau tumor suppressor gene product (pVHL), an E3 ligase that targets HIF-1α for proteasomal degradation through the recognition of an hydroxylated Pro564 [55,56,57,58,59,60].

However, whether NAA10 has significant KAT activity has been called into question by several follow-up studies. Two independent efforts found that NAA10 is not able to acetylate HIF-1α [61,62]. A 2016 paper by Magin and colleagues sowed further doubt about whether the NAA10 active site is structurally able to accommodate a lysine side chain [63]. The specific loss of KAT activity from recombinant NAA10 in a time-dependent manner, the oligomerization of purified NAA10, and NAA10 autoacetylation being a requirement for KAT activity [50,64] have been suggested as explanations for these discrepancies in the literature.

A recent study claimed that NAA10 is hydroxylated at Trp38 through factor inhibiting HIF-1α (FIH) [65]. It was suggested that HIF-1α is negatively regulated by FIH in two ways. FIH has an established role as a HIF-1α aspartate hydroxylase and acts on the Asn803 of the HIF-1α C-terminal transactivation domain, the hydroxylation of which blocks interaction with the transcriptional coactivator p300 [66,67]. The other mode of regulation was through the following proposed mechanism: the hydroxylation by the FIH of NAA10-Trp38 induces a shift in the β6–β7 loop, widening the substrate acceptor site of NAA10 and enabling the accommodation of a lysine as a NAA10 substrate. Thus, NAA10 hydroxylation was suggested to be a prerequisite for HIF-1α acetylation followed by recognition by the pVHL E3 ligase and ultimately, the proteasomal degradation of HIF-1α [65].

We aimed to quantify the extent of NAA10 Trp38 hydroxylation to define the role of this modification as a general switch for NAA10 activity and to characterize the relationship between NAA10 and FIH. However, we confidently and exclusively detected the previously reported NAA10-Trp38 hydroxylation site in the unmodified form across a panel of human cell lines. We also could not detect interaction between NAA10 and FIH nor any effect of FIH on NAA10’s enzymatic activity.

## 2. Results

### 2.1. NAA10 Is Not Hydroxylated at Trp38

To determine the degree of hydroxylation on the tTrp38 of NAA10 (NAA10-Trp38-OH), we analyzed various liquid chromatography coupled mass spectrometry (LC/MS) runs (Appendix A) where the NAA10 or NAA15 components had been enriched. The samples (*n* = 12) were derived from either HEK293, HeLa, or HAP1 cells, and the peptides had been prepared by either the filter-aided sample preparation (FASP) or in-solution digestion methods. One of the samples was prepared by NAA10-V5 immunoprecipitation (IP) in HEK293 cells and further fractionation off-line prior to LC/MS to obtain comprehensive NAA10-derived peptide coverage. The purpose was to detect NAA10 hydroxylated at the Trp38 site and to determine whether this modification could be found in different cell types. NAA10 was detected in every sample, with a combined unique sequence coverage of 97%,and ranging from 56.2% to 92.8% sequence coverage in the individual samples (Appendix A). Trp38 is contained within the 22 amino acid residue (30)YYFYHGLSWPQLSYIAEDENGK(51) tryptic peptide (herein referred to as YYF peptide). This peptide had a combined log_2_ intensity of 28.47 and was found in 10 out of 12 samples (Appendix A). However, we did not find any evidence of hydroxylation at Trp38 in any of the samples (Figure 1A, Appendix A). Kang and colleagues reported two peptide forms containing Trp38: the YYFYHGLSWPQLSYIAEDENGK peptide and a longer, 30 amino acid residue peptide resulting from a missed cleavage at Lys51, (30)YYFYHGLSWPQLSYIAEDENGKIVGYVLAK(59). We only identified the missed cleavage peptide (30)YYFYHGLSWPQLSYIAEDENGKIVGYVLAK(59) in the fractionated sample; this peptide was likewise not hydroxylated on Trp38. In addition to Trp38, Kang et al. found a YYF peptide that was hydroxylated at Asp47 and Asn49. These three different sites give rise to eight possible hydroxylation configurations, all of which were detected by Kang et al., with the exception of the triply hydroxylated peptide. We thus performed our database search with Asp-OH, Asn-OH, and Trp-OH as variable modifications. We did not find any modified version of the YYF peptide, neither the short nor the long version. NAA10 was somewhat oxidized on other Asp, Asn, and Met residues, however (Appendix A).

Based on MS/MS counts, Kang and colleagues estimated the fraction of NAA10-Trp38-OH to be 26% [65]. However, we were unable to detect any evidence of this modification. Across our dataset, we detected 58,225 peptides, with MS/MS counts ranging from 1 to 904. We had a cumulative MS/MS count of 35 for the short YYF peptide form. The longer peptide was detected once, in the fractionated sample. A MS/MS count of 35 is higher than 96.9% of the peptides in the dataset (Figure 1C). We estimated how many Trp38-OH peptides we should have found if we assumed a site occupancy of 26%. To do this, we calculated the binomial probability distribution of finding k hydroxylated peptides (*k* = number of successful trials) in 35 trials (*n*) given a probability of *p* = 0.26 (Equation (1)).
(1)$$P\left(B\;=\;k\right)\;=\;\left(\begin{array}{c} n\\ k \end{array}\right)p^{k}{\left(1-p\right)}^{n-k}$$

The likeliest outcome would have been to identify the hydroxylated peptide nine times, and the probability of finding it between two and fifteen times is 99%. The probability of not finding it at all would be less than 0.003%. Thus, we concluded that the original estimate of 26% site occupancy is unlikely to hold in our cell-based systems corresponding to HeLa, HAP1, and HEK-293. Our data conclusively show that the stoichiometry of hydroxylation at this site is very low, no more than fractions of a percent at most.

### 2.2. NAA10 Does Not Interact with FIH

Further, it was claimed that NAA10 and FIH interact. In the NAA10 IP samples analyzed by mass spectrometry, we did not find any FIH peptides (Appendix A). Note that FIH has been previously shown to be expressed in HEK293 [65,68], HeLa [68], and HAP1 cells [69]. We performed immunoblotting for FIH in each cell type and verified that this was the case (Figure 2A). To enable comparisons between LC/MS runs collected under differing conditions, we ranked protein groups using an intensity-based absolute quantification (iBAQ [70]) value for each sample and found the percentile score for each of the NatA complex subunits (Figure 2B). For the NAA10- and NAA15-enriched samples, NAA10 and NAA15 were usually in the top 10% most abundant proteins. This included the reanalysis of the LC/MS runs from a previous paper describing the NAA15-T406Y and NAA15-L814P variants, with the T406Y mutant having a significantly reduced binding to NAA50, and the L814P mutant being somewhat deficient in binding HYPK [71]. We found no FIH peptides at all in our dataset even though we could find the three other known NatA subunits for both the NAA10 and NAA15 IPs. We thus concluded that NAA10 and FIH do not form a stable complex.

### 2.3. FIH Does Not Influence the Activity of NAA10

To assess whether or not FIH impacts NAA10 catalytic activity, we performed NAA10 IP in HAP1-WT and HAP1-FIH-KO cells to obtain an endogenous NAA10 and NatA complex with and without the presence of FIH. These immunoprecipitates were used in a ^14^C-based acetylation assay [72] with synthetic peptides. A peptide with an EEEI *N*-terminus is derived from γ-actin and is a preferred in vitro substrate of NAA10 [42,43] while SESS, which is derived from high-mobility group protein A1, is a preferred NatA substrate [43]. To test any shift towards KAT activity as a function of FIH presence, we included a peptide with the sequence Ac-SGRGKGGKGLGKGGAKR, which is derived from the *N*-terminus of histone H4 and contains four lysines that can accept an acetyl group. The immunoprecipitates were analyzed by Western blotting to verify the co-IP of NAA15 with NAA10 on the beads used for the acetylation assays and to normalize the Nt-acetylation activity to the enzyme amount (Figure 3A). This blot confirmed that the NAA10 IP enriched for NAA15 but not for FIH, supporting the conclusion that NAA10 and FIH do not interact (Figure 2). High activity was observed for NAA10 immunoprecipitates towards the SESS peptide, indicating intact NatA activity. The activity of immunoprecipitated NAA10 towards EEEIA represents monomeric NAA10 activity and/or trace activity from NatA, and in our assay, it was approximately 30% higher than it was against the Nt-acetylated H4 peptide. For the Nt-acetylated histone H4 peptide, the background activity in the absence of NAA10 was considerably higher than it was for the peptides with free *N*-termini. This likely represents the nonenzymatic lysine acetylation of the histone peptide. We found no significant differences between NatA or NAA10 activity in WT versus FIH-KO cells. We have thus concluded that FIH does not affect the activity of NAA10 in or out of the NatA complex.

## 3. Discussion

Through Nt-acetylation, NAA10 makes its mark on approximately 40% of the proteome. Nt-acetylation generally has been implicated in protein stability regulation, subcellular localization, complex formation, and crosstalk with other PTMs at the *N*-terminus. Phenotypes in patients harboring pathogenic NAA10 variants have been described for several loci in the last decade [10,73,74,75,76,77,78,79,80,81,82,83,84]. Patients with NAA10-related syndromes invariably have intellectual disabilities and developmental delays, while cardiac arrhythmias, dysmorphic features, and hypotonia are also commonly seen, and there is considerable phenotypic variability [85]. These mutations can affect NAA10 catalytic activity, NAA10 stability, or NatA complex formation [1]. The importance of NAA10 is further underscored by the severe knockdown or knockout phenotypes in model organisms [44,86,87,88,89,90] or in cultured cells [10,91]. However, our understanding of the precise mechanistic contributions of NAA10 substrates to disease phenotypes is currently poor.

The biochemical function of NAA10 has been a matter of considerable debate. Its role as a NAT was established early, while its acetyltransferase-independent role as an adaptor for a DNA methyltransferase [44,45] is a more recent discovery. The role of NAA10 as a lysine acetyltransferase is controversial [63], and in particular, its activity towards HIF-1α [52] has been questioned [61,62,92]. A recent article [65] described a novel modification on NAA10, namely hydroxylation at Trp38, which proposed a novel mechanism of the NAA10 substrate specificity modulation that allows it to accept HIF-1α as a substrate, and this was the motivation for the current study. We set out to quantify the extent of this modification but were unable to detect its presence using high-resolution mass spectrometry in several samples collected over a range of conditions. Additionally, we showed that even if it is present, it is almost certainly much lower than the 26% site occupancy that has been reported. We cannot exclude the possibility that NAA10 is hydroxylated at this site under certain conditions, but if it was a widespread modification present under normoxic conditions, we would expect it to be discoverable by mass spectrometry in cells cultured in normoxia and potentially be present in other cell types as well. What could account for the confidently assigned spectrum of NAA10-Trp38-OH by Kang and colleagues? We suggest that the earlier identification of NAA10 hydroxylation may stem from a sample preparation artifact introduced during the in-gel digestion procedure. This effect has been previously described for oxidation-prone amino acids generally [93] and for tryptophan specifically [94]. The non-enzymatic oxidation of the tryptophan side chain can lead to the formation of an oxindolylalanine moiety, which would give rise to the same mass shift as a hydroxylated tryptophan [94]. 

Kang and colleagues describe the first tryptophan modification by FIH. They also reported different YYF peptide species, which were hydroxylated at Trp38 and additionally at Asp47 and Asn49 and that present either singly or doubly within the same peptide sequence. We detected neither Asp47-OH nor Asn49-OH. This array of hydroxylations indicates that artefactual oxidation may have been quite extensive in their samples. As a step in serotonin biosynthesis, tryptophan is hydroxylated at C-5 by tryptophan hydroxylase [95]. The hydroxylation of Trp38 was assigned to C-2 of the tryptophan indole ring [65], but it is not clear how this assignment is justified. Their observation of a 16 Da mass shift of the YYF peptide is explained equally well by nonenzymatic oxindolylalanine formation as it is by enzymatic C-2 hydroxylation. Our study highlights the importance of validation by the scientific community. This has previously been necessary for post-translational modifications discovered by mass spectrometry [96,97] as well as for the functional relationship between NAA10 and HIF-1α [61,62,98].

Relative to the prolyl hydroxylases (PHDs), which act almost exclusively on HIF-1α [97], the asparaginyl hydroxylase FIH has a considerable number of substrates. These are mainly ankyrin repeat domain (ARD)-containing proteins. The Asn22 of OTU domain-containing ubiquitin aldehyde-binding protein 1 (OTUB1), a component of the IL1-β pathway with which FIH also interacts [99,100], is a non-ARD FIH substrate. The majority of FIH hydroxylations are of Asn substrates, but Asp [101] and His [102] hydroxylation by FIH have been described, so the requirement for Asn as an oxygen acceptor is not strict. However, a consensus sequence for the ARD substrates of FIH has been described [103,104]; it contains a Leu in the -8 position to the acceptor Asn, an acidic residue in the -2 position, and the −1 and +1 residues that flank the acceptor residue are hydrophobic. The hydroxylated Asp in ankyrinB and ankyrinR [101] and the His in tankyrase-2 [102] are both contained within the established ADR FIH consensus sequence. The Asn22 of OTUB1 is also situated in a highly similar region with a −8 Leu and a −1 Val but has a Glu in the -3 instead of the -2 position [100]. The only consensus sequence feature that is present in NAA10 is a hydrophobic Pro C-terminal to the proposed Trp oxygen acceptor. While none of these consensus sequence features are absolutely required, this does bolster the hypothesis that the hydroxylation of NAA10 at this site is artefactual.

Although Kang et al. reported that NAA10 and FIH interact, we were unable to find any evidence of this interaction. We performed several IPs enriching NAA10 or NAA15 and demonstrated that FIH is expressed in all of the the cell types used in our study (Figure 2A) but were unable to assign a single peptide to FIH through mass spectrometry (Figure 2B). IP followed by LC/MS is used extensively to identify protein interaction partners [105]. The fact that the NatA subunits NAA15, NAA50, and HYPK were frequently identified in our IPs suggests that co-IP was generally successful. Indeed, NAA15 was identified in every NAA10 IP and vice versa (Figure 2B). We consider it likely that FIH does not stably interact with NAA10 nor the NatA complex to any large degree.

Neither were we able to find any evidence that FIH modulates NAA10 activity. This was claimed by Kang et al. to be a novel mechanism for NAA10 activity modulation, widening the substrate acceptor site of NAA10 and shifting its specificity towards accepting lysine substrates. We tested three different peptide substrates—one canonical NatA substrate, one in vitro NAA10 substrate, and an *N*-terminally acetylated histone 4-derived peptide containing several lysines that could accept an acetyl group. Immunoprecipitated NAA10 from HAP1-WT and *FIH*-KO cells were able to acetylate canonical NatA and NAA10 substrates to the same degree, but lysine acetylation of a histone 4-derived, Nt-acetylated peptide was not higher than the control. This was unaffected by the presence of FIH.

Our analyses cannot exclude the possibility that NAA10 is hydroxylated in HEK293 cells under different conditions than the ones tested here. We also cannot completely discount the possibility that NAA10 interacts with FIH under certain conditions. Indeed, NAA10 was found to interact with FIH in an earlier screen for FIH substrates [106]. This interaction was detected after treatment with the prolyl hydroxylase inhibitor dimethyloxalylglycine (DMOG). Such a substrate trapping approach might have yielded a detectable NAA10/FIH interaction in our dataset, but we note that Kang et al. detected the interaction without DMOG treatment. Nevertheless, we cannot exclude the possibility that different experimental conditions, including IP tags and washing regimen, may contribute to the differing results between the previous study and this.

In summary, we demonstrated that NAA10 is not hydroxylated at Trp38, that it does not bind FIH, and that its activity is not regulated by FIH. FIH is thus unlikely to represent a switch steering the potential KAT activity of NAA10.

## 4. Materials and Methods

### 4.1. Plasmids

Plasmids encoding NAA10-V5 [39], NAA15-V5 [39], NAA15-T406Y-V5 [71], NAA15-L814P-V5 [71], and FIH-V5 [100] have all been previously described. NAA10-ΔS205-V5 was generated from NAA10-V5 using the Q5 Site-Directed Mutagenesis Kit (New England Biolabs, Ipswich, MA, USA) and the mutagenesis primers 5′-GGTGGGGACAGCAAGGACC and 5′-ATCCTCGGCAGCCAGGCC. pcDNA3.1-V5-His-EMPTY (Invitrogen, Waltham, MA, USA) was used as a transfection control.

### 4.2. Cell Culture and Transfection

HEK293 (ATCC: CRL-3216) and HeLa (ATCC: CCL-2) cells were cultured in 10 cm dishes with 10 mL Dulbecco’s Modified Eagle Medium (Sigma) containing 10% fetal bovine serum (FBS), 3% L-glutamine, and 1% PenStrep at 37 °C and 5% CO_2_. When reaching confluency, the cells were washed with 1× PBS and were detached by trypsinization at 37 °C for 5–10 min. Detached cells were resuspended in appropriate amounts of fresh medium and were usually split 1:10 to new 10 cm dishes for further subculturing. HAP1 WT (Horizon Discovery, Cambridge, UK, catalog number C631) and *FIH*-KO (Horizon, HZGHC007289c007) cells were cultured in 10 cm dishes with 10 mL Iscove’s Modified Dulbecco’s Medium (Gibco, Waltham, MA, USA) containing 10% FBS and PenStrep (100 U/mL penicillin, 100 µg/mL streptomycin). At 50–60% confluency, the cells were washed and detached by trypsinization as previously described. Detached cells were resuspended in appropriate amounts of fresh medium and were split 1:20 to new 10 cm dishes for further subculturing. Both HAP1 cell lines were passaged until diploidy was confirmed using flow cytometry, as described in [107]. The cells were transfected at approximately 70% confluency. cells were seeded at a ratio of 1:3 24 h prior to infection. The transfection mixture was prepared by diluting DNA 1:3 in XtremeGene9 transfection reagent (Roche, Basel, Switzerland). After 15 min of incubation at room temperature, the cells were transfected through the dropwise addition of 500 µL of the transfection mix per 10 cm dish. The growth medium was replaced 24 h post transfection. Cells were harvested 48 h post transfection in 1400 µL cold PBS using a cell scraper, and the cells were centrifuged at 4 °C at 1000× *g*. The pellets were then resuspended in IPH lysis buffer (50 mM Tris-HCl pH 8.0, 150 mM NaCl, 5 mM EDTA, 0.5% NP-40) supplemented with cOmplete EDTA-free protease inhibitor cocktail (Roche) and were incubated on ice for 30 min. The lysate was cleared at 17,000× *g* for 5 min before IP or SDS-PAGE analysis.

### 4.3. Immunoprecipitation for Mass Spectrometry

In order to enrich NAA10-V5, NAA15-V5, or NAA15 and any bound proteins, IP was performed in lysate from cells transfected with NAA10-V5, NAA15-V5, or untransfected cells. For the samples analyzed by 3 h LC/MS methods, cell lysates were incubated with 2–2.5 µg antibody (for specific antibodies, see Appendix A) per 10 cm dish at 4 °C on a rotating wheel for 2–3 h. After preparing 30 µL Dynabeads Protein G (Pierce, Waltham, MA, USA) per 10 cm dish by washing them three times in 1 mL IPH buffer, they were added to each sample. The antibody complexes were bound to the beads during an overnight incubation at 4 °C on a rotating wheel. The next day, the beads were isolated using a magnetic holder, washed three times in 1 mL IPH buffer, and resuspended in 100 µL 1× sample buffer followed by boiling for five minutes. For the sample labeled “frac”, 10 dishes of HEK293 cells (10 cm) were transfected at approximately 80% confluence with 5 µg NAA10-V5 (pTA01) for 48 h and were pooled before lysis and IP. The other samples were prepared in biological triplicates (for the remaining in-solution digested samples) or without biological replicates (the remaining samples prepared by FASP). The cells were harvested with trypsin, washed twice with PBS, and kept on ice for all subsequent steps. The cells were lysed in IPH buffer with 0.5% NP-40, and the lysate was used for IP with 2 µg anti-V5 and 20 µL Dynabeads Protein G magnetic beads per 10 cm cell dish. The beads were preloaded with anti-V5 for 20 min and were washed twice in IPH buffer. IP proceeded at 4 °C for 1 h (for the samples prepared by in-solution digestion) or overnight (the samples prepared by FASP). The beads were washed three times with IPH buffer and, for the samples to be prepared by in-solution digestion, they were washed a further three times with detergent-free IPH buffer to remove detergents that could interfere with downstream MS analysis.

### 4.4. Sample Preparation for Mass Spectrometry

Some samples (Appendix A) were prepared using the FASP protocol, essentially as previously described [108]. Samples prepared by in-solution digestion were prepared in the following manner: the beads were resuspended in a preheated elution buffer (6 M guanidine-HCl, 5 mM TCEP, 10 mM chloroacetamide, 100 mM Tris-HCl (pH 8.5)) containing 5 ng/µL LysC and were incubated on a shaker for 1 h at 25 °C. The supernatant was transferred to a new tube, and the beads were washed once with 250 µL 25 mM Tris-HCl (pH 8.5). The wash solution was combined with the eluate, and trypsin was added to a final concentration of 5 ng/µL and incubated over night at 37 °C with shaking. Digestion was stopped by adding TFA to a final concentration of 1%. Peptides were desalted on a 50 mg SEP-PAK C18 column (Waters) on a vacuum manifold. The columns were activated with 100% ACN and equilibrated three times with 0.1% TFA (all volumes 1 mL). The sample was loaded and washed three times with 0.1% TFA and was eluted by 250 µL 40% ACN/0.1% TFA and then by 250 µL 60% ACN/0.1% TFA. Peptides were dried in a Speedvac and were resuspended in 5% ACN/0.1% formic acid prior to LC/MS analysis. One sample (Frac) was additionally fractionated by the high-pH StageTip method, a modification of the original StageTip protocol [109,110]. After drying in a Speedvac, the peptides were resuspended in 100 µL 50 mM ammonium bicarbonate. All StageTips steps were performed into an autosampler plate in a swing-bucket centrifuge at room temperature and with 300× *g* with 50 µL volumes. A self-packed StageTip with four C18 disks was activated with 100% methanol and was equilibrated three times with 50 mM ABC. The sample was loaded, and the flowthrough fraction was collected. Then, elution was performed with increasing steps of ACN in 50 mM ABC (5%, 7%, 12%, 18%, 25%, 30% and 50% ACN) with each fraction collected separately. After the last elution, the plate was dried completely in the Speedvac, and peptides were reconstituted in 10 µL 5% ACN/0.1% formic acid. 

### 4.5. Western Blotting

The lysates and IP samples collected 48 h post-transfection were mixed with sample buffer, boiled for 5 min at 95 °C, and run on a Mini-PROTEAN TGX Stain-Free Precast Gel (Bio-Rad, Hercules, CA, USA). PageRuler Plus Prestained Protein Ladder (Thermo Fisher Scientific, Waltham, MA, USA) was used as a molecular weight marker, and the gels were run in TGS electrophoresis buffer (25 mM Tris, 192 mM glycine, 0.1% SDS, pH 8.3) at 100 V for 5 min and then at 200 V for 40 min. For Western blotting, proteins were transferred in TG buffer (25 mM Tris, 192 mM glycine, pH 8.3) at 100 V for 40 min, and the blotting was confirmed by Ponceau staining. The membranes were blocked in 5% dry milk in PBS and 0.1% Tween (PBST) at room temperature for 1 h with gentle shaking before they were incubated with primary antibody (Appendix A) in 1% dry milk over night at 4 °C on a rocking shaker. The membrane was rinsed for unbound antibody by washing three times with PBST for 5 min prior to 1 h incubation with secondary antibody (Appendix A) in 3% dry milk at RT. The membrane was washed with 1× PBST three times and with PBS once for 5 min and was developed using SuperSignal West Pico Chemiluminescent Substrate (Thermo Fisher Scientific). Imaging of the membrane was performed using a ChemiDoc XRS+ (Bio-Rad). Visualization was performed in ImageLab™ v. 6.0.1 (Bio-Rad). 

### 4.6. ^14^C Acetylation Assays

Immunoprecipitated NAA10 and NatA complexes were used in an in vitro ^14^C-Ac-CoA-based acetylation assay for measuring the catalytic activity. ^14^C-Ac-CoA was used as a donor upon the acetylation of peptides by NAA10, and product formation could be detected and quantitated due to radioactivity. A P18 phosphocellulose filter quenched the enzymatic reaction through the binding of positively charged peptides, and excess ^14^C-Ac-CoA was washed off. By covering the filter in scintillation fluid, energy from the β-particles was absorbed, which resulted in photons being emitted. The acetylated peptides could be quantified based on the light emitted, and the measured activity was then adjusted to the amount of NatA or NAA10 used in the reactions, which was determined from Western blotting analysis. Even low levels of acetylation could be detected with this highly sensitive method. ^14^C-acetylation assays were performed as described [72] and were performed pairwise for the FIH KO and WT HAP1 cells in order to compare the catalytic activity of immunoprecipitated NatA and NAA10 as well as to evaluate the potential NAA10 KAT activity towards an Nt-acetylated histone 4 peptide. The NAA10 antibody was used for the NAA10 IPs, while control IPs were performed with a rabbit IgG control antibody (Appendix A). Dynabeads with immunoprecipitated NAA10 were washed three times in IPH lysis buffer and were resuspended in 125 µL acetylation buffer (100 mM Tris-HCl pH 8.5, 2 mM EDTA, 20% glycerol). Three replicates with 200 μM peptide (SESS, EEEI or Nt-acetylated histone 4, all substrate peptides were synthesized by Biogenes) (Appendix A), 50 μM ^14^C-labelled Ac-CoA, 10 μL IP beads, and dH_2_O were mixed to a final volume of 25 μL. Two negative replicates without a peptide were also prepared. All of the samples were incubated at 37 °C and 1300 rpm on a thermoshaker for 30 min, with the exception of the SESS samples, which were incubated for 15 min. After incubation, the magnetic beads were isolated using a magnetic holder, and a 23 μL sample was transferred to P81 phosphocellulose filter squares. The filter squares were washed for 3 × 5 min in 10 mM HEPES buffer (pH 7.4) followed by air drying on paper. The dried filter squares were placed in individual tubes and were soaked in 5 mL scintillation fluid for 14C-signal measurement using a scintillation counter (Tri-Carb 2900TR Liquid Scintillation Analyzer, PerkinElmer, Waltham, MA, USA). The IP samples were further analyzed by Western blotting, and the measured activity was adjusted to quantification of corresponding anti-NAA10 and anti-NAA15 bands. Bands were quantified in Image Lab 6.0.1 (Bio-Rad) relative to the wildtype bands for each antibody.

### 4.7. LC/MS

LC/MS samples were run using one of two different methods. Samples prepared by the FASP method were injected into an Ultimate 3000 RSLC system (Thermo Fisher Scientific) connected online to a QExactive HF mass spectrometer (Thermo Fisher Scientific) equipped with EASY-spray nano-electrospray ion source (Thermo Fisher Scientific) by PROBE. The samples were loaded and desalted on a pre-column packed with 3 µm C18 beads (Acclaim PepMap 100, 2 cm × 75 µm ID nanoViper column, Thermo Fisher Scientific) at a flow rate of 5 µL/min for 5 min with 0.1% trifluoroacetic acid. The mass spectrometer was operated in the DDA-mode to automatically switch between full scan MS and MS/MS acquisition. Instrument control was through Qexactive HF Tune 2.9, and Xcalibur 4.1. MS spectra ranging from *m*/*z* 375–1500 were acquired in the Orbitrap with a resolution of 120,000 at *m*/*z* 200, an automatic gain control (AGC) target of 3 × 10^6^, and a maximum injection time of 100 ms. The 12 most intense eluting peptides (Top12 method) above intensity threshold 50,000 counts and charge states 2 to 5 were sequentially isolated to a target value (AGC) of 10^5^ and a maximum injection time of 110 ms in the C-trap, with the isolation width maintained at 1.6 *m*/*z* (offset of 0.3 *m*/*z*) before fragmentation in the Higher-Energy Collision Dissociation cell. Fragmentation was performed with a normalized collision energy of 28%, and fragments were detected in the Orbitrap at a resolution of 30,000 at *m*/*z* 200, with first mass fixed at *m*/*z* 100. One MS/MS spectrum of a precursor mass was allowed before the dynamic exclusion for 25 s with “exclude isotopes” on. Lock-mass internal calibration (*m*/*z* 445.12003) was used. The spray and ion-source parameters were as follows: ion spray voltage = 1800 V, no sheath and auxiliary gas flow, and capillary temperature = 275 °C.

For the samples prepared by in-solution digestion, including the high-pH stagetip fractionated sample, each fraction was run in 30 min gradients on a QExactive-HF X mass spectrometer (Thermo Fisher Scientific) equipped with an in-house packed 15 cm, 75 µm ID capillary column with 1.9 µm Reprosil-Pur C18 beads (Dr. Maisch, Ammerbuch-Entringen, Germany), as described [111].

### 4.8. LC/MS Data Analysis and Processing

LC/MS data were searched in Maxquant (version 1.6.17.0) using the integrated search engine Andromeda [112,113] against a human proteome database retrieved from Uniprot on 13 October 2020 containing 20,360 protein canonical sequences. A decoy database consisting of reverse sequences from this database was automatically generated by the search engine. Protease specificity was set to specific trypsin cleavage; that is, cleavage C-terminally to lysine or arginine, except when followed by a proline. A maximum of two missed cleavages were permitted. Methionine oxidation and hydroxylation of tryptophan, aspartic acid, and asparagine were set as variable modifications, and the carbamidomethylation of cysteine as a fixed modification. Peptide-spectrum matches and site and protein false discovery rates were set to 0.01. Label-free quantification (LFQ), iBAQ, and matches between runs and dependent peptides were all enabled. During the search, technical replicates of each sample were treated as a single experimental group in Maxquant; consequently, there are between one and three raw files per experiment. Maxquant output tables peptides.txt, modificationSpecificPeptides.txt, and proteinGroups.txt were analyzed in Perseus (version 1.6.5.0) [114]. Proteins and peptides identified by reverse sequences were removed, as were the protein groups that were only able to be identified by a modified peptide or by a single peptide. To enable comparisons of the relative abundance of the YYF peptide between diverse experiments, protein groups were ordered by log_10_ iBAQ and peptides by log_2_ intensity and were expressed as percentile scores within each sample. MS/MS spectra identified by Maxquant were annotated and visualized with seeMs [115]. The MS proteomics data have been deposited to the ProteomeXchange Consortium via the PRIDE partner repository with the data set identifier PXD023655.

## Figures and Tables

**Figure 1 ijms-22-11805-f001:**
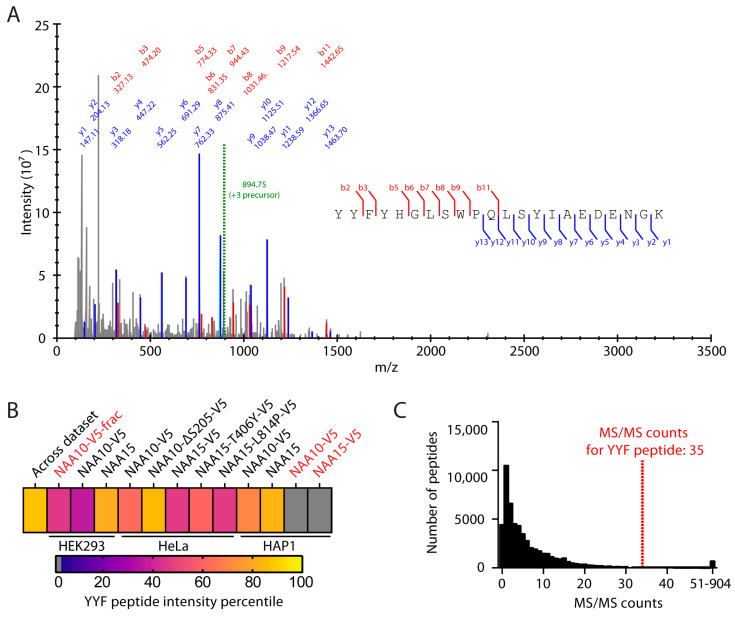
NAA10 is not hydroxylated at Trp38. (**A**) MS/MS spectrum of YYF peptide identified by MaxQuant and annotated in SeeMS. Green line: mass of precursor. Red: b ions. Blue: y ions. (**B**) Intensity percentile of YYF peptide in each sample analyzed by LC/MS. Red text: sample prepared by in-solution digestion. Black text: sample prepared by FASP. (**C**) Histogram of MS/MS counts among the identified peptides across the dataset.

**Figure 2 ijms-22-11805-f002:**
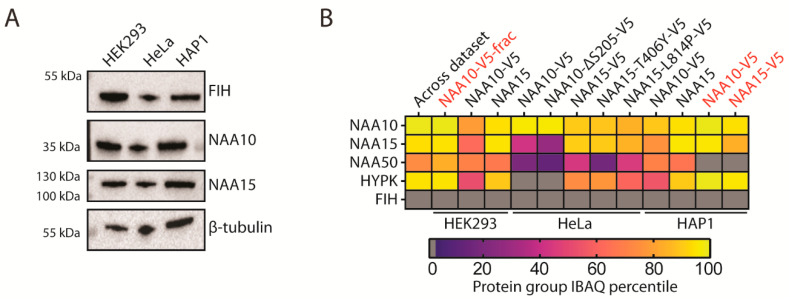
NAA10 does not interact with FIH. (**A**) Immunoblots of HEK293, HeLa, and HAP1 cells with the indicated antibodies. (**B**) Percentile score of iBAQ values for the known NatA complex subunits and FIH in each sample. Red text: samples prepared by in-solution digestion. Black text: samples prepared by FASP.

**Figure 3 ijms-22-11805-f003:**
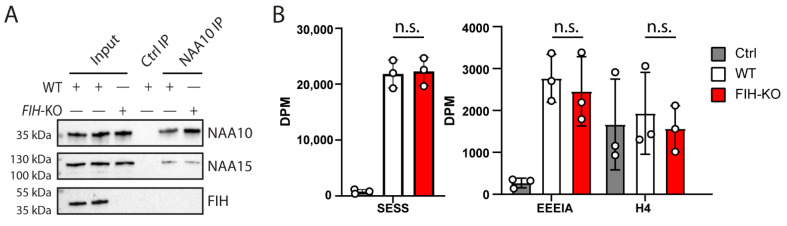
FIH does not influence the enzymatic activity of NAA10. (**A**) Immunoblots of the NAA10 IPs from HAP1-WT and HAP1-*FIH*-KO cells. (**B**) NatA and monomeric NAA10 activity measurement from ^14^C-acetylation assay of immunoprecipitated NAA10 and coprecipitated NAA15. The activity towards SESS was normalized to NAA15, and the activity towards EEEI and histone 4 peptide was normalized to NAA10 to show relative product formation per enzyme. n.s., not significant (one-way ANOVA with Šídák correction for multiple testing).

## Data Availability

The MS proteomics data have been deposited to the ProteomeXchange Consortium via the PRIDE partner repository with the data set identifier PXD023655.

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
