# Peer review of "Hydroxylation of the Acetyltransferase NAA10 Trp38 Is Not an Enzyme-Switch in Human Cells"

_ijms, 2021, doi:10.3390/ijms222111805_

Round 1

Reviewer 1 Report

Proteins possess N-terminal amino acid signals (N-degrons) and most of the human proteome is subject to N-terminal acetylation (Nt) catalyzed by N-terminal acetyl-transferases (NATs). The origin of the degrons is not yet clear, but their functional effects are measurable. The catalytic and functional consequences of NAA10 variants are sometimes different and, therefore, it is necessary to understand the different cellular roles of NAA10 in physiological and/or pathological mechanisms.

The authors report that "the KAT activity of NAA10 towards hypoxia-inducible factor 1α (HIF-1α) was recently found to depend on the hydroxylation at Trp38 of NAA10 by factor inhibiting HIF-1α (FIH)". In their conclusions, the authors claim that they "could not detect hydroxylation of Trp38 of NAA10 and found no evidence that NAA10 interacts with or is regulated by FIH". Many articles report the lysine acetyltransferase (KAT) activity of NAA10. Thus, we cannot have doubts about the acetylating action of the enzyme. Other investigations also suggest that the enzyme could not acetylate HIF. In this respect, many "explanations for these discrepancies" can be found in the literature.

The Authors of this manuscript prove their assertion with an excellent experimental design and adequate methodologies. But an inevitable observation calls into question both sides. The expression, and therefore the activity, of an enzyme, is strongly space and time-dependent. In order to understand its role, it is necessary to know with great precision the where, when, and how it operates. The lack of this crucial information turns any functional observation into an in vitro experiment, outside of time and space. This means that we will find appreciable quantities of a protein, or its covalent modification, only if we operate in the right space-time context. Also, in this case, we could fail if the absolute values of the protein molecules are very low or with a very short average life. However, enrichment techniques, or even kinetic traps, can help, but such an experimental effort takes much longer, and/or sophisticated tools.

Circadian rhythms are at the heart of homeostatic control because they organize the tissue- and organism-wide metabolism in time and space. The circadian clock performs the daily variations in the metabolic homeostasis of mammals, allowing the temporal coordination of their physiology. The network of clocks in peripheral tissues interacts through cross-interactions to drive much of the circadian gene expression and respond to tissue-extrinsic signals. Therefore, the circadian rhythm controls the fluctuations in the expression of many genes involved in important specific metabolic activities (for instance, glycidic, lipid, and energy metabolism in the liver) with an impact on the activity of the transcriptional cycle, down to the protein level. Hence, many proteins are integrated into a circadian proteome. NAA10 "catalyzes the co-translational N-terminal (Nt-) acetylation of 40% of the human proteome", so the vastness of its catalytic action must certainly be integrated into much of the circadian proteome. This should be the correct context.

If we analyze the work of Joo-Wong Jeong et al., we can see that these researchers used the yeast two-hybrid system to identify candidate proteins that interact with HIF-1α in vivo. This method is very effective in finding physical binding between proteins, but this does not mean that between proteins there is also a functional relationship. Even the GST pull-down assay, which they then used, is another in vitro technique to detect physical interactions between two or more proteins as a tool for confirming a predicted protein-protein interaction or identifying novel interacting partners.

Even all the analyzes made later on the regulatory relations of this protein do not include the use of its activity but only of its structural properties. In vitro functional tests, too, do not prove a functional interaction in vivo because they are an experimental forcing out of time and space. The same consideration applies to experiments on cellular model systems. With these approaches, we do not prove what really happens in a functioning organism, nor do we know which molecular species are functionally active. We only badly mimic the feasibility of the event, since, when a protein is appropriately functioning in the tissues, it is adapted to the "where" by its gene expression, to how by post-transcriptional modifications, which also define its turnover time.

Considering the publication date, they did their best. Today there are single-cell technologies that can give much more precise and detailed information, but they are still little used and people continue to speculate as they did thirty years ago.

I suggest the authors change their discussion by adding these methodological aspects as a new discrepancy. I also suggest introducing the biological reasons that cause it.

Reviewer 2 Report

The authors of the work "Hydroxylation of the acetyltransferase NAA10 Trp38 is not an enzyme-switch in human cells" have developed a very exciting, but at the same time, complex work. Proving a negative result is a very difficult task, but it is also part of the very essence of science (the basic pillar of refutability).
They have addressed a series of premises and have tried to obtain independent results that, according to their conclusions, contradict what was previously stated by other researchers. I have to make a series of comments, because to reinforce your results there can be no doubts (perhaps they only arise for me, but they would also need to be resolved in case it happens to others).

First, some easy things of style or shape:
1) The alignment of the results and discussion paragraphs are not justified.
2) Line 239, "their sample (s)".
3) Line 398, H2O (needs subscript)

Later,
4) Were the samples prepared (and therefore the results shown) with biological replicates?
5) In "Cell culture and transfection", do the authors perform a check on the efficiency of the transfection? Any resistance marker that allows to select only the transfected cells?
6) The wording of "Immunoprecipitation for mass spectrometry" is a bit confusing. Although in other sections the authors follow the same format (first explain the technique and then give details of how they have done it), this section could be simplified to make it clearer.
7) Are controls used in the "Immunoprecipitation for mass spectrometry" assay? The efficiency of the immunoprecipitation is checked? the fact that NAA10 and NAA15 are, only "usually" in the top 10 could be indicating that many other things are being coprecipitated. In Figure 3A an immunoprecipitation control is shown, in consisting of?
8) Are the immunoprecipitation gels shown? They should be able to be observed to verify the efficiency of the technique, as well as possible non-specific retentions (which, in turn, could have some kind of effect on other possible interactions)
9) The authors say (lines 388 and 389) that they use Western blotting analysis to determine the amount of NatA or NAA10 used in the reactions, did they make a Western with known concentrations of those proteins to obtain a calibration line with which to interpolate (since that would be the way to obtain the amounts that would be used?
10) Are the authors trying to demonstrate the non-formation of a stable complex? Could it be that what was stated by other researchers refers to the formation of a complex that is not so stable as to withstand the experimental conditions of the present work?
11) What does the scale in figure 2B indicate?
